#  High *KYNU* Expression Is Associated with Poor Prognosis, *KEAP1*/*STK11* Mutations, and Immunosuppressive Metabolism in Patient-Derived but Not Murine Lung Adenocarcinomas

**DOI:** 10.3390/cancers17101681

**Published:** 2025-05-16

**Authors:** Ling Cai, Thomas J. Rogers, Reza Mousavi Jafarabad, Hieu Vu, Chendong Yang, Nicole Novaresi, Ana Galán-Cobo, Luc Girard, Edwin J. Ostrin, Johannes F. Fahrmann, Jiyeon Kim, John V. Heymach, Kathryn A. O’Donnell, Guanghua Xiao, Yang Xie, Ralph J. DeBerardinis, John D. Minna

**Affiliations:** 1Children’s Research Institute, University of Texas Southwestern Medical Center, Dallas, TX 75390, USAralph.deberardinis@utsouthwestern.edu (R.J.D.); 2Quantitative Biomedical Research Center, Peter O’Donnell Jr. School of Public Health, University of Texas Southwestern Medical Center, Dallas, TX 75390, USA; reza.mousavijafarabad@utsouthwestern.edu (R.M.J.); guanghua.xiao@utsouthwestern.edu (G.X.); yang.xie@utsouthwestern.edu (Y.X.); 3Simmons Comprehensive Cancer Center, University of Texas Southwestern Medical Center, Dallas, TX 75390, USA; kathryn.odonnell@utsouthwestern.edu (K.A.O.); john.minna@utsouthwestern.edu (J.D.M.); 4Department of Molecular Biology, University of Texas Southwestern Medical Center, Dallas, TX 75390, USA; 5Department of Thoracic and Head and Neck Medical Oncology, University of Texas MD Anderson Cancer Center, Houston, TX 77030, USA; agalan@mdanderson.org (A.G.-C.);; 6Hamon Center for Therapeutic Oncology Research, University of Texas Southwestern Medical Center, Dallas, TX 75390, USA; 7Department of Pharmacology, University of Texas Southwestern Medical Center, Dallas, TX 75390, USA; 8Department of General Internal Medicine, University of Texas MD Anderson Cancer Center, Houston, TX 77030, USA; 9Department of Clinical Cancer Prevention, University of Texas MD Anderson Cancer Center, Houston, TX 77030, USA; 10Departments of Urology and Cellular & Molecular Physiology, Yale School of Medicine, New Haven, CT 06510, USA; 11Hamon Center for Regenerative Science and Medicine, University of Texas Southwestern Medical Center, Dallas, TX 75390, USA; 12Department of Bioinformatics, University of Texas Southwestern Medical Center, Dallas, TX 75390, USA; 13Howard Hughes Medical Institute, University of Texas Southwestern Medical Center, Dallas, TX 75390, USA; 14Department of Internal Medicine, University of Texas Southwestern Medical Center, Dallas, TX 75390, USA

**Keywords:** lung adenocarcinoma, kynurenine pathway, tryptophan catabolism, NAD metabolism, kynureninase, KYNU, immune suppression, KEAP1, STK11, prognostic biomarker, mouse model limitations

## Abstract

Lung adenocarcinoma is a common and deadly form of lung cancer, and better tools are needed to predict how aggressive the disease will be and which treatments may work best. In this study, we focused on a metabolic enzyme called kynureninase (KYNU), which showed two distinct patterns of activity across lung tumors. We found that high KYNU activity often signals a poor prognosis and is linked to specific genetic changes in cancer cells. Interestingly, KYNU appears to affect how tumors interact with the immune system and how they process certain nutrients. Our findings also suggest that current mouse models may not accurately reflect how KYNU works in human lung tumors, which is important for designing better therapies. By understanding KYNU’s role, we hope to guide future research in cancer metabolism and improve treatment strategies.

## 1. Introduction

Lung cancer remains the leading cause of cancer-related deaths worldwide, with lung adenocarcinoma (LUAD) being the most common subtype, accounting for 40% of cases [1]. Advances in early diagnosis, staging, and multimodal treatments—including surgery, radiotherapy, and chemotherapy—have expanded LUAD treatment options. The identification of oncogenic drivers, development of targeted therapies, and integration of immune checkpoint blockade (ICB) and antibody-based therapies have further improved survival in many LUAD molecular subsets. However, significant challenges remain, including the need for robust biomarkers to guide the selection of optimal systemic therapies for individual patients, improving the long-term durable responses to both ICB and targeted therapies, preventing metastases, and overcoming resistance, which frequently occurs in all classes of systemic therapy [1]. In lung cancer clinical translational research, many therapeutic advances have been driven by the identification of a potential biomarker (e.g., mutated *EGFR* and *KRAS*) and their validation in patient-derived preclinical models (e.g., cell lines, xenografts) and mouse models (e.g., genetically engineered mouse models, GEMMs). These models have been instrumental in demonstrating the functional significance of biomarkers by showing that targeting them produces an anti-tumor effect. Hence, assessing the functional significance of biomarkers and their compatibility with traditional syngeneic and patient-derived (xenograft) mouse models is essential for facilitating clinical translatability.

Over the past decade, large-scale gene expression studies have advanced biomarker discovery by linking oncogenotypes to gene expression programs and patient outcomes. Just as LUADs can have “bimodal” oncogenotypes (e.g., presence or absence of *TP53*, *KRAS*, or *EGFR* mutations) that influence their pathogenesis, pathophysiology, and response to treatment, genes exhibiting dramatic bimodal expression patterns (very high or very low) may also play a key role in LUAD biology and therapeutic response [2]. Leveraging our Lung Cancer Explorer (LCE, http://lce.biohpc.swmed.edu/) [3], an integrated resource of expression and clinical data from >6700 patients across 56 studies, we applied Gaussian mixture modeling to identify genes exhibiting bimodal expression patterns [4]. These distributions suggest potential oncogenotype-driven gene reprogramming, revealing genes that may serve as strong prognostic markers and have functional significance.

Here, we identify *KYNU* expression, which encodes a metabolic enzyme in the kynurenine pathway, as a bimodally distributed prognostic biomarker in LUAD. Through a meta-analysis of 23 LUAD datasets, we show that bimodal stratification of *KYNU* expression more accurately captures prognosis compared with median-based dichotomization. High *KYNU* expression correlates strongly with *KEAP1/STK11* co-mutations in LUADs, enhancing the precision of prognostic models beyond mutation status alone. Our metabolomics analysis of LUAD cell lines revealed a distinct KYNU-associated metabolic profile, including elevated anthranilic acid, increased NADP, and depletion of niacinamide. Furthermore, we identify a translational gap in murine preclinical models: KYNU’s biology cannot be effectively modeled in traditional genetically engineered mouse systems, limiting its preclinical functional study. Finally, extending our computational analysis to other cancers, we find high *KYNU* expression is also associated with worse outcomes in many cancer types, including pancreatic cancer, but is associated with better prognosis in melanoma. These findings highlight the importance of cancer lineage-specific considerations when developing therapeutic strategies targeting the kynurenine pathway.

## 2. Materials and Methods

### 2.1. Determination of Bimodal Distribution

We implemented Gaussian mixture model clustering with two pre-defined clusters and equal variances using the “Mclust” function from the R package mclust [4]. The bimodal index was computed using the “bimodalIndex” function from the R package BimodalIndex, which fits data to a two-component mixture model with equal variance.

### 2.2. Meta-Analysis of Lung Adenocarcinoma Studies for Survival Association with Gene Expression

The collection and reprocessing of data from lung cancer studies with both gene expression and clinical data was described previously [3]. Among 56 available datasets, we selected 23 datasets that included both overall survival data and tumor gene expression data for LUAD patients. Each dataset contained at least 10 patients, totaling 3114 cases. For each gene, we performed survival association analysis using two approaches: (1) Bimodal clustering-based dichotomization, comparing survival between high and low expression groups, and (2) Continuous gene expression analysis, using sample-wise z-score standardized values. We used a univariate Cox proportional hazard regression model to assess the association between gene expression and overall survival, estimating hazard ratios and *p*-values for each dataset. A random effects meta-analysis was then performed to estimate summary hazard ratios and *p*-values across datasets.

### 2.3. Processing of Gene Expression Data

Transcript per million (TPM) RNA-seq data for CCLE [5] were downloaded from the DepMap portal under version 19Q1 (https://depmap.org/portal/download/). Only cell lines with “tcga_code” annotated as “LUAD” from the CCLE annotation table were included for analyses. Gene-normalized RNA-seq data from TCGA LUAD, processed by RSEM (RNA-Seq by Expectation-Maximization) algorithm, were downloaded from Firebrowse (http://firebrowse.org/) (doi:10.7908/C19P30S6). FPKM RNA-seq data for CPTAC were downloaded from dbGaP with accession number phs001287 and retrieval date 3 June 2020. Data preprocessing included quantile normalization and/or log transformation, applied as appropriate.

### 2.4. Processing of Mutation Data

Mutation data for CCLE were downloaded from the DepMap portal as “CCLE_DepMap_18q3_maf_20180718.txt”. Mutation data for TCGA were retrieved from Firebrowse on 14 September 2017 by running the “Analyses.Mutation.MAF” function from the R package “FirebrowseR” [6]. For both CCLE and TCGA, mutations with variant annotations such as “Silent”, “Intron”, or “IGR” were removed. Mutation data for CPTAC were downloaded from dbGaP with accession number phs001287 with a retrieval date of 3 June 2020. Mutations annotated as “LOW” in the “IMPACT” field or have a non-empty “GDC_FILTER” were removed. For downstream analyses, we used the union of mutations detected by at least one of the following variant callers: MuTect, MuSE, PinDel, Sniper, or VarScan.

### 2.5. Screen for Oncogenotype Associated with High KYNU Expression

The Cancer Gene Census (CGC) catalog of genes implicated in cancer was retrieved from the COSMIC website (https://cancer.sanger.ac.uk/census) on 2 September 2019. In each LUAD mutation dataset (CCLE, TCGA, and CPTAC), we analyzed only CGC-listed genes with mutations detected in at least 9 independent samples. A Mann–Whitney U test was used to compare *KYNU* expression between mutants and wild-type samples for each gene. We applied Benjamini–Hochberg procedures for multiple comparisons but focused on nominal *p*-values, given the relatively small sample size.

### 2.6. Determination of KEAP1/STK11 Status in Different Datasets

In addition to mutation data, we incorporated additional molecular datasets to assess the functional status of *STK11* and *KEAP1* across our study cohorts. For CCLE, gene fusion, translocation, and copy number data were downloaded as “CCLE_Fusions_20181130.txt”, “CCLE_translocations_SvABA_20181221.xlsx”, and “public_19Q1_gene_cn.csv” files, respectively, from the DepMap portal. Translocation or fusion events involving *STK11* or *KEAP1* at the breakpoint were classified as “loss” for the tumor suppressor gene status. Additionally, copy number values smaller than −1 were also indicative of loss. Since LKB1 protein levels showed a bimodal distribution in reverse phase protein array (RPPA) data, we performed model-based clustering to classify the cell lines. Those with LKB1 RPPA levels below the model-determined threshold were annotated as LKB1 loss. For TCGA, “TCGA_genomic_alterations.tsv” was downloaded from cBioPortal [7]. We classified mutations as loss events if they were not annotated as “AMP” (amplification), “not profiled”, or “no alteration”. For CPTAC, we relied solely on mutation data to define loss status without incorporating additional molecular data.

### 2.7. Pathway Analyses for KEAP1/STK11 Associated Gene Expression

Gene set enrichment analysis (GSEA) [8] was used for three sets of pathway analyses comparing genes differentially expressed in samples with or without mutations in *STK11*, *KEAP1,* or both. The GSEA R script was adapted from the original script downloaded from MSigDB [8] (https://www.gsea-msigdb.org/gsea/downloads.jsp). The signal-to-noise ratio was used as the gene ranking metric. Gene ontology terms of biological processes (also from MSigDB) were used as the input geneset library. Results from CCLE, TCGA, and CPTAC datasets were reviewed. To identify genes consistently upregulated in both *KEAP1*/*STK11* co-mutants and samples with mutations in *KEAP1*-only or *STK11*-only, we first identified gene sets with nominal *p*-values below 0.05 and positive normalized enrichment scores (NES) across all three datasets. From each of the “GO_POLYKETIDE_METABOLIC_PROCESS” (shared between “KEAP1mutSTK11mut-KEAP1wtSTK11wt” and “KEAP1mutSTK11wt-KEAP1wtSTK11wt” results) and “GO_CAMP_CATABOLIC_PROCESS” (shared between “KEAP1mutSTK11mut-KEAP1wtSTK11wt” and “KEAP1wtSTK11mut-KEAP1wtSTK11wt” results) genesets, genes appear in multiple leading-edge lists were selected for heatmap visualization. Similarly, for genes downregulated in *KEAP1*/*STK11* co-mutants in TCGA and CPTAC (but not CCLE), the most frequently appearing genes from the leading-edge list of geneset “GO_MHC_PROTEIN_COMPLEX_ASSEMBLY” were used in the heatmap. The most frequent macrophage-related genes appeared in the leading-edge lists from two gene sets: “GO_MACROPHAGE_COLONY_STIMULATING_FACTOR_SIGNALING_PATHWAY” and “GO_RESPONSE_TO_MACROPHAGE_COLONY_STIMULATING_FACTOR”. Enrichment plots were generated using modified functions from the R package “fgsea” [9] to overlay results across datasets. The *p*-values were empirically estimated using permutated sample labels [8].

### 2.8. Tumor Immune Infiltrate Association Analyses

Estimates of TCGA tumor immune infiltrates, generated using different computational algorithms by Li et al. [10], were downloaded from the supplementary table available on the publisher’s website. TCGA LUAD tumor samples were classified into *KYNU*-high and *KYNU*-low groups, and the association between *KYNU* expression and immune infiltrate estimates was assessed by Pearson correlation within each group. For each algorithm, we examined gene signatures from the original publication or software repository to exclude signatures containing *KYNU*. We identified that *KYNU* was part of the monocyte lineage signature in MCPcounter and, therefore, excluded MCPcounter scores from our analyses [11].

### 2.9. Pathway Analysis for KYNU-Associated Genes

In TCGA and CPTAC datasets, we assessed the correlation between *KYNU* expression and the expression levels of all other genes in the genome using Pearson correlation. This analysis was performed for all samples in each dataset, a subgroup of samples without *KEAP* or *STK11* mutations and the remaining group of samples with *KEAP1* and/or *STK11* mutations. Multiple comparison adjusted *p*-values were calculated by Benjamini–Hochberg procedures. Genes with adjusted *p*-values less than 0.05 in both the TCGA and CPTAC datasets were selected for pathway enrichment analysis. A hypergeometric test was applied to identify enriched canonical pathways using gene sets from MSigDB [8]. Multiple comparison-adjusted *p*-values were further generated for the pathway *p*-values.

### 2.10. Analysis of scRNA-Seq Data from Healthy Human Lung

The Travaglini_2020 dataset consists of single-cell RNA sequencing (scRNA-seq) data from healthy human lung tissue [12]. Processed count data and cell type annotations provided by the authors were downloaded from Synapse (Accession ID: syn21041850). For this study, FACS-sorted SmartSeq2 data were used. Cell types with fewer than 10 cells were excluded from analyses. Library size normalization was performed using the “library.size.normalize” function from the R package “phateR” [13]. *Log*_2_-transformed expression data were used for visualization. Cell types in the figure were ordered based on their average *KYNU* expression.

### 2.11. Molecular Features Associated with KEAP1/STK11 Oncogenotypes or KYNU Expression

Metabolomics data from CCLE [5] were downloaded from the DepMap portal as “CCLE_metabolomics_20190502.csv”. CPTAC proteomics data were extracted from a supplementary table of the original CPTAC LUAD paper [14]. To assess molecular associations, we applied Pearson correlation or one-way ANOVA, depending on the data type and distribution.

### 2.12. Pan-Cancer Survival Analyses

TCGA pan-cancer survival data were downloaded from the supplementary table “NIHMS978596-supplement-1.xlsx” in Liu et al. [15]. TCGA pan-cancer RNA-seq data, reprocessed by Toil pipeline [16], was downloaded as “tcga_RSEM_Hugo_norm_count” from Xena (https://tcga.xenahubs.net). In addition to cancer types, sample types (primary tumor, blood, metastasis) were taken into consideration for patient stratification.

### 2.13. Other R Packages Used for Analyses

All analyses were conducted in R (version 4.2.2) [17]. The following R packages were used: data wrangling: openxlsx [18], data.table [19], dplyr [20], plyr [21], tidyverse [22], reshape2 [23], statistical analyses: stats [17], survival [24], meta [25], survminer [26] graph visualization: ggplot2 [27], GGally [28], ggridges [29], ggrepel [30], grid [17], patchwork [31], cowplot [32], highcharter [33], ComplexHeatmap [34], viridis [35], DescTools [36], RColorBrewer [37], scales [38] table visualization: finalfit [39], kableExtra [40], webshot [41], and formattable [42].

## 3. Results

### 3.1. KYNU mRNA Expression Is Bimodally Distributed in Lung Adenocarcinoma, Associated with Protein Expression, and Its High Expression Is Associated with Poor Prognosis

We analyzed gene expression from 3114 patients across 23 lung adenocarcinoma (LUAD) datasets and identified *KYNU* as a top bimodally distributed gene with high expression associated with poor prognosis (Figure 1a). Other bimodally expressed genes identified in our analyses, such as *FOXM1* [43,44], MELK [45], *RGS20* [46,47,48], and *OIP5* [49], have been previously reported to associate with clinical outcomes and play functionally significant roles in lung adenocarcinoma. Notably, while many top prognostic genes exhibit strong coexpression patterns, *KYNU* and *RGS20* expression remain largely independent, showing only weak correlations with expression of the other bimodally expressed genes. This suggests that they may contribute to LUAD pathophysiology through distinct, non-overlapping mechanisms (Appendix A). *KYNU* encodes kynureninase, an enzyme that links the tryptophan catabolism pathway to de novo NAD synthesis. Its expression in LUAD exhibits a distinct bimodal distribution (Figure 1b), and survival analysis revealed that model-based clustering dichotomization by Gaussian mixture models outperforms median-based approaches in capturing *KYNU*’s prognostic significance (Figure 1c,d; Appendix A). Importantly, using DNA molecular annotation data from 74 LUAD cell lines, we observed no correlation with *KYNU* copy number or mutations with *KYNU* mRNA expression levels (Figure 1e). Importantly, using proteomic data from CPTAC LUAD [14], we confirmed that *KYNU*’s bimodal expression at the mRNA level is highly concordant (r = 0.88) with its protein-level expression (Figure 1f).

Kynurenine, a metabolite in the tryptophan catabolism pathway, has been well-characterized as an immunosuppressive molecule [50,51], acting as an endogenous ligand for the aryl hydrocarbon receptor (AhR) [52]. AhR activation promotes a tolerogenic immune environment through multiple mechanisms [53], including driving T cell differentiation into regulatory T cells [54]. Given that KYNU enzymatically degrades kynurenine, its association with poor prognosis was unexpected, as it would seemingly counteract kynurenine’s immunosuppressive effects. However, the activation of the kynurenine pathway in cancer cells may lead to local tryptophan depletion, restricting its availability to immune cells and thereby suppressing anti-tumor immunity [55]. Additionally, increased kynureninase activity may support tumor growth by enhancing de novo synthesis of nicotinamide adenine dinucleotide (NAD), a critical cofactor in bioenergetics and redox homeostasis (Figure 1g). The complexities of the kynurenine pathway’s role in cancer pathogenesis and anti-tumor immunity are well recognized and have been extensively reviewed by Gouasmi et al. [56].

To contextualize *KYNU* within related pathways, we examined the following gene sets: “Kynurenine Metabolites Suppress T-Cells in Cancer Immune Escape” from Elsevier Pathway Collection [57], “Tryptophan Catabolism” from Reactome [58], and “Reactive Oxygen Species Pathway” from the MSigDB Hallmark database [8]. Among these pathways, *KYNU* expression stood out because of its consistent bimodal distribution across studies and its stronger association with poor prognosis compared with the expression of other pathway members (Appendix A). This suggests that *KYNU* expression is regulated in a distinct manner, potentially linking it to tumor aggressiveness.

### 3.2. KYNU Expression Is Upregulated in KEAP1/STK11 LUAD Co-Mutants

To identify genetic determinants of *KYNU* upregulation, we analyzed datasets that report both gene expression and mutation data, including CCLE [5], TCGA [59], and CPTAC [14]. Across all three datasets, *KEAP1* and *STK11* mutations were consistently associated with elevated *KYNU* expression (Appendix A).

We further examined structural variations and assessed the impact of *KRAS*, *KEAP1*, and *STK11* alterations on *KYNU* expression. *KYNU* expression was particularly elevated in cells harboring both *KEAP1* and *STK11* mutations, a combination that was well represented across the datasets (Figure 2a). Although *KRAS* frequently co-mutates with *KEAP1* and *STK11* [60], we did not observe significant *KYNU* upregulation by *KRAS* mutation status alone in two of the three datasets. Notably, *KYNU* expression was more pronounced in patient metastasis-derived cell lines with *STK11* and *KEAP1* mutations (Appendix A). The higher frequency of *KEAP1*/*STK11* co-mutations in patient-derived cell lines compared with tumor datasets (Figure 2b; Appendix A) suggests a potential selective advantage for in vitro establishment, possibly because of enhanced glutamine utilization and reactive oxidative stress resistance [61]. This may also be explained by kynurenine pathway activation, which has been implicated in promoting anchorage-independent survival, potentially supporting cell viability during in vitro replating [62].

This trend was particularly evident in a panel of isogenic HCC515-derived cell lines generated from a prior study [63]. Cells with single *KEAP1* or *STK11* mutations exhibited modest *KYNU* upregulation compared with wild-type cells, while *KEAP1* and *STK11* co-mutant cells displayed the highest *KYNU* expression. This pattern highlights a clear additive effect of the two mutations on *KYNU* regulation (Figure 2c, left). To assess whether this effect was unique to *KYNU*, we compared its expression with canonical NRF2 and LKB1 pathway targets: *NQO1* (regulated by NRF2) and *PDE4D* (regulated by LKB1). Unlike *KYNU*, *NQO1* and *PDE4D* showed unidirectional regulation: *NQO1* was elevated only in *KEAP1* mutants, while *PDE4D* was increased only in *STK11* mutants (Figure 2c, middle and right). The absence of a stepwise pattern for these genes underscores the distinct regulatory mechanism of *KYNU*, which integrates inputs from both pathways.

Beyond comparing *KYNU* expression by driver mutation status, we further assessed its expression relative to other NRF2 targets (*G6PD*, *NQO1*) and LKB1 targets (*CPS1*, *PDE4D*) across datasets. While *G6PD* and *NQO1* exhibited a strong correlation, along with *CPS1* and *PDE4D*, *KYNU* displayed only a modest correlation with these pathway genes (Appendix A). Although *G6PD* and *CPS1* also demonstrated bimodal expression patterns, survival analysis showed that their prognostic significance, when assessed using model-based clustering, was less pronounced than that of *KYNU* (Appendix A compared with Figure 1d). These findings collectively highlight *KYNU* as a shared downstream target of NRF2 and LKB1 signaling that stands out for its selective upregulation in *KEAP1*/*STK11* co-mutants, which may contribute to its strong prognostic value.

### 3.3. KYNU Expression Provides Prognostic Value Independent of KEAP1/STK11 Co-Mutations in LUAD

Previous studies have shown that co-mutations of *KRAS/STK11* or *KRAS/KEAP1* are associated with significantly shorter survival in NSCLC patients [64], with *KEAP1*/*STK11* co-mutations conferring an even worse outcome [65]. Using survival and mutation data from 1258 LUAD patients in the MSK-IMPACT dataset [66], we found that patients with co-mutations in *KEAP1/STK11* had markedly shorter survival compared with those with mutations in only one of these genes, regardless of *KRAS* status (Figure 3a–c). Patients with single or co-occurring *STK11* and *KEAP1* mutations had similar distributions in sex and primary versus metastatic tumor status compared with the mutation-negative group. However, never-smokers were markedly underrepresented among *STK11* and/or *KEAP1*-mutant patients (0.1–0.8%), whereas they constituted 23% of the mutation-negative cases (Figure 3d). To further investigate the frequency and relationship of *KEAP1* and *STK11* mutations in LUAD, we analyzed AACR GENIE data from 5709 LUAD patients. *KEAP1*/*STK11* co-mutations were present in 11% of cases, with an approximately equal distribution between *KRAS*-mutant and *KRAS*-wild-type patients (Figure 3e). Interestingly, *KEAP1*/*STK11* co-mutations occurred more frequently than *KRAS/KEAP1* or *KRAS/STK11* co-mutations, suggesting a functional synergy between NRF2- and LKB1-regulated pathways in tumor progression. To further explore the oncogenic landscape of tumors harboring *KEAP1*/*STK11* co-mutations without concurrent *KRAS* mutations, we evaluated alterations in key growth signaling pathways driven by RTKs, RAS/MAPK, or PI3K/AKT/mTOR pathway genes (Figure 3f). Among the 483 tumors with *KEAP1*/*STK11* co-mutations and wild-type *KRAS*, no known alterations in these pathways were detected in 250 tumors (>50%), indicating that *KEAP1*/*STK11* co-mutations may be sufficient to drive oncogenesis in a large subset of patients. This supports a distinct oncogenic mechanism in this subgroup, potentially involving metabolic reprogramming and stress response deregulation, independent of canonical RTK pathway activation.

Interestingly, in the TCGA LUAD cohort, we found that high *KYNU* expression predicts poor survival, regardless of *KEAP1/STK11* co-mutation status (Figure 4a,b). Specifically, among the 29 patients with *KEAP1*/*STK11* co-mutations, those with low *KYNU* expression (n = 15) exhibited much longer median survival—comparable to patients without *KEAP1*/*STK11* mutations. Conversely, 33 out of 354 patients without *KEAP1*/*STK11* mutations displaying high *KYNU* expression experienced shorter median survival (Figure 4b). To further validate the prognostic significance of *KYNU* expression, we performed a multivariate analysis controlling for *KEAP1*/*STK11* status, smoking status, age, gender, and TNM stage. *KYNU* expression remained significantly associated with worse overall survival in the TCGA LUAD cohort (Figure 4c). Validation with an independent dataset—the CPTAC LUAD cohort—using univariate analyses confirmed that high *KYNU* expression is consistently associated with poor survival outcomes, regardless of *KEAP1*/*STK11* co-mutation status (Figure 4d,e). These findings suggest that while *KYNU* expression is co-regulated by LKB1 loss and *KEAP1* mutation-mediated NRF2 activation, it provides additional prognostic value beyond the mutation status of *STK11* and *KEAP1*.

### 3.4. Tumor-Intrinsic and Microenvironmental Sources of KYNU Expression Underlying Its Bimodal Distribution in LUAD

Given previous reports linking *KEAP1/STK11* co-mutations to an immune cold phenotype in LUAD [67], we sought to explore the relationship between *KYNU* expression and tumor immune infiltration. As a first step, we examined the expression of *PTPRC* (encoding CD45), a pan-leukocyte marker, to evaluate whether total immune cell presence varied with *KYNU* expression. A biphasic relationship was observed between *PTPRC* and *KYNU* expression across multiple LUAD datasets. Specifically, a positive correlation was evident only in *KYNU*-low tumors, suggesting distinct immune microenvironments between *KYNU*-low versus *KYNU*-high tumors (Figure 5a). In TCGA LUAD *KYNU*-low tumors, immune infiltrate score estimates [10] revealed the strongest correlations between *KYNU* expression and macrophage scores (Figure 5b). Additional positive correlations were observed with neutrophils, myeloid dendritic cells, and plasmacytoid dendritic cells, suggesting that *KYNU* expression in *KYNU*-low tumors may originate from myeloid cells. This pattern was further corroborated by our re-analysis of scRNA-seq data from healthy human lung tissue [12], which confirmed high *KYNU* expression in multiple myeloid cell subsets, particularly macrophages and plasmacytoid dendritic cells (Figure 5c). In contrast, *KYNU* expression was minimal in epithelial cell types, except for goblet cells. Notably, goblet cells also exhibited high levels of *KMO*, *IDO1*, and *IDO2* expression, suggesting higher kynurenine pathway activity. We extended this analysis to two independent scRNA-seq LUAD datasets [68,69], which showed that *KYNU* expression was primarily restricted to myeloid lineage cells (Figure 5d,e).

In contrast, *KYNU*-high tumors exhibited an inverse relationship between *KYNU* expression and immune infiltrate estimates in TCGA LUAD (Appendix A), suggesting reduced immune infiltration. To further investigate the molecular landscape, we performed gene set enrichment analysis (GSEA) comparing *KEAP1*/*STK11* co-mutants (double mutants), single mutant, and wild-type LUAD samples. The results revealed that rather than forming a completely distinct signature, double mutants exhibit a convergence of metabolic gene programs observed in single mutants, particularly those involved in tumor-intrinsic processes such as cAMP catabolism and polyketide metabolism (Appendix A). Notably, MHC II-related antigen presentation pathways, which are associated with the tumor microenvironment (TME), were significantly downregulated in double mutants (Appendix A). Furthermore, macrophage-related gene expression was uniformly low in both single and double mutants, supporting the conclusion that *KYNU* expression in *KYNU*-high tumors originates from cancer cells rather than infiltrating immune cells (Appendix A, black box).

Together, these findings demonstrate that *KYNU* expression represents two distinct biological contexts in LUAD. In *KYNU*-low tumors, expression is primarily driven by myeloid cell infiltration, whereas in *KYNU*-high tumors, it predominantly reflects expression by cancer cells. This duality underscores the potential of *KYNU* as a biomarker for stratifying immune microenvironments in LUAD and guiding tailored therapeutic strategies. Given the observed dichotomy in KYNU expression contexts, we next explored whether these distinct transcriptional profiles are associated with underlying metabolic differences.

### 3.5. KYNU Metabolomics Association in Patient-Derived LUAD Lines Identifies a Compensatory Metabolic Mechanism That Provides a Basis for a LUAD Immune Suppressive Microenvironment

With the discovery that high *KYNU* expression correlates with *KEAP1*/*STK11* mutations, we recognized a paradox. On the one hand, kynurenine is a well-known immunosuppressive metabolite that should be degraded by high *KYNU* expression [54]. On the other hand, *STK11* and *KEAP1* mutations are strongly associated with an immunosuppressive tumor microenvironment [60,67]. Consistent with this paradox, our analysis revealed an inverse correlation between *KYNU* expression and immune infiltrates in *KYNU*-high tumor samples (Figure 6a). To explore this further, we examined CCLE metabolomics data [70] from 72 LUAD cell lines to identify metabolic profiles associated with *KYNU* upregulation (Appendix A). We identified several metabolites correlated with *KYNU* expression as a result of *KEAP1* or *STK11* mutations, including reduced glutathione and 1-methylnicotinamide (Appendix A). The increase in glutathione is likely driven by *KEAP1* loss (Appendix A), as NRF2 activation induces the expression of genes encoding the catalytic (GCLC) and modifier (GCLM) subunits of glutamate-cysteine ligase, the rate-limiting enzyme in GSH synthesis [71]. Similarly, the increase in 1-methylnicotinamide could be attributed to the *STK11* mutations (Appendix A), as LKB1 regulates the enzyme Nicotinamide N-methyltransferase (*NNMT*), which catalyzes this metabolic conversion [72]. We have validated these relationships in both CCLE RNA-seq data and in CPTAC LUAD tumor proteomics data (Appendix A).

Notably, anthranilic acid, the product of kynureninase, exhibited the strongest positive correlation with *KYNU* RNA expression (Figure 6a). While its levels varied significantly across *KEAP1/STK11* oncogenotypes (Figure 6b), a multivariate linear model controlling for *KEAP1/STK11* mutations confirmed that *KYNU* expression independently drives anthranilic acid upregulation (*p*-value = 0.0029) (Figure 6c). In contrast, kynurenine, the substrate of kynureninase, showed no correlation with *KYNU* expression (Figure 6d–f). The absence of kynurenine depletion, despite increased kynureninase activity, suggests a compensatory mechanism that sustains kynurenine levels, possibly through enhanced kynurenine synthesis. This maintenance of kynurenine levels in the face of high *KYNU* expression, coupled with anthranilic acid upregulation, provides a basis for immune suppression in the LUAD tumor microenvironment.

Beyond anthranilic acid, we identified additional metabolites associated with *KYNU* expression. NADP levels exhibited a positive correlation (r = 0.41, *p* = 3 × 10^−4^), while niacinamide (nicotinamide) levels showed a negative correlation (r = −0.3, *p* = 1 × 10^−2^ (Figure 6g–l). The upregulation of NADP may be influenced by the activation of de novo NAD synthesis via the kynurenine pathway. Increased NADP availability could enhance redox homeostasis and anabolic metabolism, supporting tumor survival. In contrast, niacinamide depletion with higher KYNU expression suggests a reduced reliance on NAD salvage pathways. Since niacinamide is a key precursor for NAD synthesis in the salvage pathway, its depletion may indicate a metabolic shift favoring de novo NAD biosynthesis from tryptophan-derived metabolites rather than the recycling of niacinamide. Together, these findings suggest that *KYNU*-driven metabolic reprogramming alters NAD metabolism, which also potentially favors pathways that support tumor adaptation to stress and immune evasion.

### 3.6. Potential Translational Challenges in Using Genetically Engineered Mouse Models of LUAD to Study the Role of KYNU in Lung Cancer Pathogenesis and Therapy Development

Selecting appropriate preclinical models is critical for uncovering the functional role of *KYNU* in LUAD. While human data demonstrate a gradual increase in *KYNU* expression in tumors with *KEAP1* or *STK11* mutations, with the highest levels observed in co-mutated tumors, translating these findings into genetically engineered murine models remains a challenge. The abundance of *KYNU* in myeloid cells suggests a potential role in immune regulation. An obvious question is the tumor cell-autonomous role of KYNU expression in LUAD growth and survival. Importantly, in patient-derived LUAD cell lines, *KYNU* expression showed no correlation with sensitivity to CRISPR- or RNAi-mediated depletion of *KYNU* in DepMap data [73]. In fact, in DepMap, only 3/1178 human cancer cell lines showed *KYNU* dropout in CRISPR screens, and a significant gene effect was only found in endometrioid ovarian cancer. These findings highlight the need for in vivo LUAD models to explore KYNU’s interactions with the TME and its broader immune-modulatory roles.

To assess the feasibility of syngeneic models, we analyzed scRNA-seq data from LKR13 syngeneic mouse models, which include *Kras*, *Kras/Keap1*, *Kras/Lkb1*, and *Kras/Keap1/Lkb1* mutant tumors [67]. Across all genotypes, *Kynu* expression was absent (Figure 7a). Further examination of autochthonous tumors from genetically engineered mouse models (GEMMs) using previously published data [63] also revealed a discrepancy with patient tumors (Figure 7b). In these models, *Kynu* expression was highest in tumors wild-type for *Keap1*/*Stk11* and decreased in *Keap1* and/or *Stk11* mutants, contradicting the patterns observed in human LUAD. Given the previously noted reduction in macrophage-related gene expression in *KEAP1* and/or *STK11* mutant patient tumors, these findings suggest that *Kynu* expression in GEMMs is primarily driven by macrophage infiltration rather than cancer cell-intrinsic expression.

We extended our analysis to pan-cancer syngeneic mouse model data [74] and consistently observed low *Kynu* expression in tumors across all tested models (Figure 7c). Conversely, *Kynu* was highly expressed in immune-rich tissues, such as the spleen and lymph nodes, reinforcing the notion that *Kynu* expression in mice is primarily immune cell-derived. However, in CCLE data of human patient-derived cancer cell lines, many non-LUAD cancer cell lines exhibited robust cell-intrinsic *KYNU* expression (Figure 7d). In these cell lines, *KYNU* was upregulated even in the absence of *KEAP1/STK11* co-mutation, indicating that *KYNU* regulation varies across cancer lineages.

Collectively, our data indicate a lack of robust, cancer cell-intrinsic *Kynu* expression in mouse models, highlighting a species-specific difference that poses challenges for preclinical modeling of KYNU’s role in LUAD using syngeneic mouse models.

### 3.7. Cancer Lineage-Specific Prognostic Considerations for KYNU and Kynurenine Pathway

Expanding beyond LUAD, we investigated *KYNU*’s prognostic associations across additional cancer types (Figure 8a). Notably, high *KYNU* expression was linked to better outcomes in two melanoma subtypes: uveal melanoma (UVM) and metastatic skin cutaneous melanoma lesions (SKCM.met) (Figure 8b). This association parallels the failure of IDO inhibitor trials in melanoma, prompting us to investigate the prognostic impact of *IDO1* expression across pan-cancer datasets. Strikingly, *IDO1*, along with other kynurenine pathway genes previously targeted in melanoma (*IDO2* and *TDO2*), also correlated with better outcomes in SKCM.met (Figure 8c,d). These findings suggest that failed clinical therapeutic targeting of the kynurenine pathway in melanoma may have an explanation, as its components appear to be associated with favorable prognoses in this context.

Conversely, in many cancer types, similar to findings in LUAD, high *KYNU* expression was associated with worse outcomes. Notably, kidney chromophobe cancer (KICH), pancreatic adenocarcinoma (PAAD), and thymoma (THYM) exhibited distinct bimodal *KYNU* expression distributions (Figure 8e). The separation between high- and low-*KYNU* groups in these tumor lineages was particularly striking, with hazard ratios exceeding 10. This dramatic outcome divergence underscores *KYNU*’s prognostic significance in these cancers and suggests its potential as a biomarker for aggressive disease phenotypes, particularly in PAAD. Taken together, these findings highlight the complexity of *KYNU*’s prognostic role, which varies across different cancer types.

## 4. Discussion

Kynurenine, a key metabolite in the tryptophan catabolism pathway, is well known for its immunosuppressive effects, including the suppression of T-cell proliferation and effector functions [75]. This has driven therapeutic strategies aimed at inhibiting upstream enzymes or delivering exogenous kynureninase to deplete kynurenine levels [76]. However, our findings highlight an apparent paradox: high *KYNU* expression, which facilitates kynurenine breakdown, correlates with poor prognosis in LUAD. This contradiction suggests that kynureninase’s role extends beyond simply reducing kynurenine-driven immunosuppression. Similarly, Fahrmann et al. and León-Letelier et al. reported that elevated *KYNU*, regulated by NRF2 activation, is linked to an immunosuppressive tumor microenvironment and poor outcomes across multiple cancer types [77,78]. For a comprehensive overview of therapeutic strategies targeting individual enzymes in the kynurenine pathway, refer to the recent review by León-Letelier et al. [79]. These findings support the hypothesis that kynureninase’s biological impact on cancer is multifaceted, potentially involving metabolic reprogramming and immune evasion mechanisms.

In LUAD, *STK11* and *KEAP1* mutations have been implicated in promoting aggressive tumor behavior [63,67]. Our data reveal that *KYNU* expression is highest in LUAD tumors with *KEAP1/STK11* co-mutations, reflecting an independent yet incremental regulation by the NRF2 and *STK11*/LKB1 pathways. This pattern is distinct from canonical NRF2 or LKB1 targets, which are regulated primarily by a single pathway. For example, *NQO1* is specifically elevated in *KEAP1* mutants, while *PDE4D* is upregulated in *STK11* mutants. The unique regulation of *KYNU* suggests it serves as an integrative node in metabolic pathways governed by NRF2 and LKB1. While NRF2 activation promotes redox homeostasis by enhancing antioxidant responses and NAD biosynthesis, *STK11*/LKB1 loss disrupts mitochondrial function and elevates reactive oxygen species (ROS), creating an increased demand for NAD to counteract oxidative stress. By linking tryptophan catabolism to de novo NAD synthesis, kynureninase may help reconcile opposing metabolic pressures, ensuring cancer cell survival under conditions of redox and bioenergetic stress. This dual regulation underscores kynureninase’s pivotal role in shaping the metabolic landscape of aggressive LUAD. In addition to the association of high *KYNU* expression and *KEAP1*/*STK11* mutations, the association between *KYNU* expression and poor prognosis remains significant even after controlling for *KEAP1/STK11* mutation status and other clinical variables, reinforcing its independent prognostic value.

Our findings reveal that *KYNU* expression is upregulated in LUAD, yet kynurenine levels, which should be degraded by KYNU, in fact, remain stable, suggesting a compensatory mechanism that sustains immunosuppressive signaling despite increased *KYNU* expression. In contrast, anthranilic acid, the direct product of kynureninase, exhibited the strongest positive correlation with *KYNU* expression in LUADs, further supporting kynureninase enzymatic activity in these tumors. Additionally, we observed a shift in NAD metabolism, with higher NADP and lower niacinamide levels, suggesting a preference for de novo NAD biosynthesis over salvage pathways. This metabolic shift may enhance redox balance and metabolic flexibility in tumors.

A major limitation of our current study is that our findings are based on correlative analyses from LUAD cell line metabolomic and transcriptomic data rather than direct manipulation of *STK11*, *KEAP1*, or *KYNU* in controlled experimental settings. Future studies involving genetic or pharmacologic perturbation of these pathways in isogenic models will be essential to establish causal relationships and assess the therapeutic potential of targeting kynureninase-associated metabolic shifts. Additionally, while our analysis focused on anthranilic acid–the product of kynurenine cleavage by KYNU, the available CCLE metabolomics dataset did not measure 3-HK or 3-HAA. Thus, although our findings suggest active kynurenine-to-anthranilate conversion, we cannot exclude the possibility that KYNU also modulates alternative branches of the kynurenine pathway in lung cancer cells. Furthermore, while kynurenine is a known activator of AHR signaling, it is important to recognize that kynurenic acid–another potent AHR ligand, can be generated independently of kynurenine via IL4I1-mediated oxidation of tryptophan [80]. IL4I1 activity has been implicated in shaping immunosuppressive tumor microenvironments [81], suggesting that integrating IL41 expression and kynurenic acid measurements into future analyses may provide additional insights into the broader landscape of tryptophan metabolism and immune remodeling in LUAD. Of course, another major limitation and subject of future research is the direct demonstration that high *KYNU* expression is associated with similar metabolomic changes in LUADs in patients.

A major challenge in understanding kynureninase’s role in LUAD is the translational gap between human and murine models. Our analyses indicate that *KYNU* expression in murine tumors does not recapitulate the oncogenotype-driven, cancer cell-intrinsic expression observed in humans. Instead, in murine tumor models, *Kynu* is primarily expressed in myeloid cells, highlighting species-specific differences in its regulation. Our human scRNA-seq analysis revealed high *KYNU* expression in goblet cells (Figure 5c), which are specialized epithelial cells responsible for mucus secretion in the respiratory tract. This raises the question of whether LUAD that express high levels of *KYNU* have some connection to or derivation from goblet cells. While abundant in the human lung, goblet cells are notably rare in the mouse lung [82]. Furthermore, goblet cells increase in response to irritants such as cigarette smoke, leading to mucus hypersecretion and potential airway obstruction [83]. Given that *KEAP1*/*STK11* co-mutated tumors are extremely rare in never-smokers (Figure 3d), it is possible that goblet cell hyperplasia induced by tobacco exposure plays an important role in the carcinogenesis of this subtype. Notably, the Laughney et al. dataset includes three tumors with *KEAP1*/*STK11* co-mutations [69]. However, no significant increase in *KYNU* expression was observed in these co-mutant tumors, and the fraction of epithelial cells was relatively small (10%). This could indicate a technical limitation of the scRNA-seq workflow, where cancer cells with high mucin expression could be difficult to dissociate, potentially leading to an underrepresentation of *KYNU*-high epithelial cells. Supporting this possibility, a previous KYNU immunohistochemistry (IHC) study in LUAD tumors confirmed *KYNU* expression in cancer cells [78], suggesting that *KYNU*-high tumor cells may be selectively lost or underrepresented in single-cell dissociation protocols.

Our analysis revealed distinct biological contexts of *KYNU* expression in LUAD. In *KYNU*-low tumors, expression is predominantly myeloid-derived, correlating with higher immune infiltration, including macrophages and dendritic cells. Conversely, *KYNU*-high tumors exhibit cancer cell-intrinsic expression and a depleted immune microenvironment, highlighting kynureninase’s dual role as a marker for stratifying immune contexts in LUAD. Interestingly, 3-hydroxyanthranilic acid (3-HAA), a kynureninase product derived from both 3-hydroxy kynurenine (3-HK) and anthranilic acid (see Figure 1g for pathway), inhibits nitric oxide synthase (NOS) expression in macrophages [84], suggesting that under normal conditions, kynureninase may act as a feedback regulator, tempering macrophage activity to maintain immune balance. However, in cancer, this mechanism may be hijacked. Elevated *KYNU* expression in cancer cells could lead to excessive 3-HAA production, suppressing NOS expression and reactive nitrogen species (RNS) generation. This suppression could impair macrophage effector functions and contribute to an immunosuppressive tumor microenvironment. Additionally, de novo NAD synthesis from the kynurenine pathway plays a pivotal role in macrophage effector responses [85]. Kynureninase-driven depletion of NAD precursors may disrupt these responses, further compromising macrophage function. These dual impacts—on RNS generation and NAD metabolism—highlight kynureninase’s potential as a critical regulator of immune evasion in LUAD and underscore the therapeutic promise of targeting *KYNU* to restore immune function in the tumor microenvironment.

Beyond LUAD, *KYNU*’s prognostic significance varies strikingly across cancer types. For example, in melanoma, high *KYNU* expression is paradoxically associated with better outcomes, aligning with the failure of IDO inhibitor trials [86]. This suggests that prior clinical efforts targeting the kynurenine pathway may have focused on the wrong cancer type. In melanoma, where enzymes in the pathway appear protective, their role may not be representative of other cancers. In contrast, in cancers such as pancreatic adenocarcinoma (PAAD) and thymoma (THYM), *KYNU*-high tumors are associated with significantly worse survival, with some hazard ratios exceeding 10. The stark survival disparities between *KYNU*-high and *KYNU*-low groups in these cancers underscore the considerable untapped potential of targeting *KYNU* and the kynurenine pathway, particularly in contexts where they clearly drive tumor aggressiveness.

## 5. Conclusions

Our study reveals that *KYNU* is a robust, bimodally distributed prognostic biomarker in LUAD, with elevated expression strongly associated with *KEAP1*/*STK11* co-mutations, reduced immune infiltration, and poor clinical outcomes. Metabolomic analyses suggest that high *KYNU* expression corresponds with active kynurenine-to-anthranilate conversion and altered NAD metabolism. Notably, we find that common murine LUAD models fail to recapitulate the cancer cell-intrinsic *KYNU* expression pattern observed in patients, underscoring a critical translational gap. These findings support the need for improved preclinical systems and highlight kynureninase and its downstream metabolites as potential therapeutic targets in LUAD and other kynureninase-driven cancers.

## Figures and Tables

**Figure 1 cancers-17-01681-f001:**
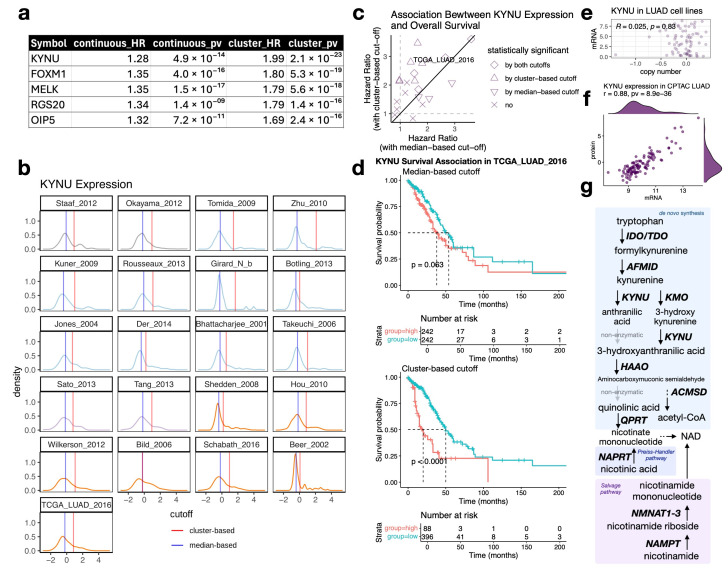
*KYNU* is bimodally distributed and associated with a worse prognosis in LUAD. (**a**) Top five genes identified by Cox proportional hazards (CoxPH) analysis with greater significance when dichotomized by model-based clustering compared with using continuous gene expression. Hazard ratios (HR) and *p*-values are shown. (**b**) Each subplot shows the distribution of *KYNU* expression (x-axis) for a specific LUAD dataset. The x-axis represents sample-wise z-transformed expression values of *KYNU* across samples, and the y-axis represents the density estimate (probability density) of those values. Cutoff values were determined using either model-based clustering (red) or the median expression level (blue). In all but one dataset, the cluster-based cutoff was higher than the median-based cutoff, suggesting that *KYNU* expression is positively skewed across these datasets. (**c**) Comparison of hazard ratios from median-based vs. cluster-based dichotomization across multiple LUAD studies. Model-based clustering yielded larger hazard ratios and more statistically significant results than median-based dichotomization, highlighting its enhanced ability to capture survival differences linked to *KYNU* expression. (**d**) Kaplan–Meier survival curves for *KYNU*-high vs. *KYNU*-low TCGA LUAD patients using cutoffs determined by either the median or model-based clustering. The *p*-values were determined by log-rank tests. (**e**) *KYNU* copy number expression does not correlate with *KYNU* mRNA expression in LUAD cell lines. (**f**) KYNU protein levels strongly correlate with mRNA expression in the CPTAC LUAD dataset. A bimodal distribution was also observed at the protein levels. (**g**) Schematic representation of the kynurenine pathway and its role in NAD Synthesis.

**Figure 2 cancers-17-01681-f002:**
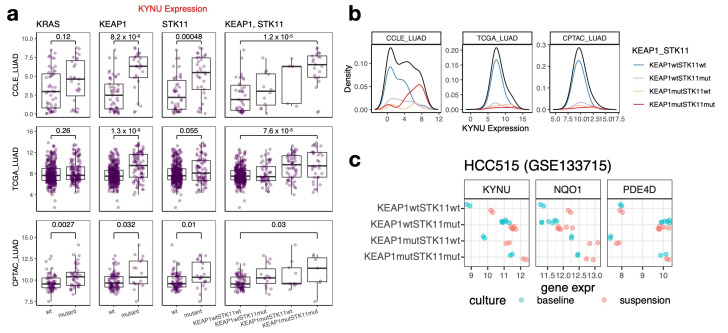
Association between *KYNU* expression and mutation status of *KRAS*, *KEAP1*, *STK11*, and *KEAP1/STK11* co-mutation in LUAD datasets. (**a**) KYNU expression in LUAD datasets (CCLE, TCGA, and CPTAC) stratified by the mutation status of KRAS, KEAP1, STK11, and KEAP1/STK11 co-mutations. The *p*-values for group comparisons were calculated using the Wilcoxon rank-sum test. *KYNU* expression is significantly elevated in *KEAP1* and *STK11* mutants, with the highest expression observed in co-mutants. (**b**) Distribution of *KYNU* expression by *KEAP1*/*STK11* co-mutation status across CCLE, TCGA, and CPTAC datasets. A wider separation and higher expression levels are noted in co-mutants within the CCLE dataset compared with TCGA and CPTAC. (**c**) Stepwise upregulation of *KYNU* expression is observed from single mutants to double mutants (*KEAP1*/*STK11* co-mutations). In contrast, *NQO1*, an NRF2-regulated gene, is predominantly upregulated by *KEAP1* mutations but not *STK11*. *PDE4D*, a gene suppressed by LKB1, is upregulated specifically in the *STK11* mutant background. Both *KYNU* and *NQO1* expression are further increased when cells transition from adherent culture to suspension culture, consistent with NRF2 pathway activation under suspension conditions.

**Figure 3 cancers-17-01681-f003:**
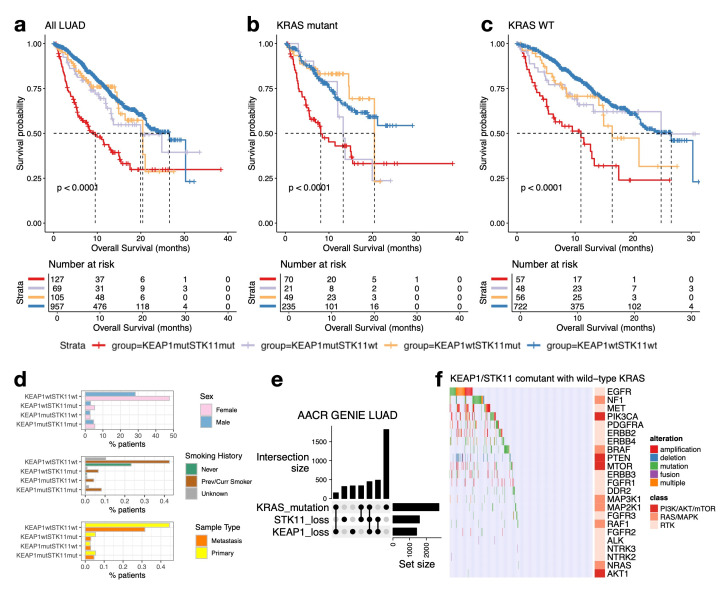
Clinical characteristics of *KEAP1/STK11* co-mutated tumors (**a**–**c**). Kaplan–Meier survival plots of MSK-IMPACT LUAD cohort stratified by *KEAP1*/*STK11* mutation status in (**a**) all patients, (**b**) *KRAS* mutant patients, and (**c**) *KRAS* wild-type (WT) patients. (**d**) *KEAP1* and *STK11* oncogenotypes according to sex, smoking status, and primary versus metastasis status. (**e**) UpSet plot illustrating the intersections for *KRAS*, *KEAP1*, and *STK11* alterations from the AACR GENIE LUAD cohort. The intersection bar chart highlights the frequencies of combinatorial mutation patterns, revealing frequent co-mutation of *KEAP1* and *STK11* with or without *KRAS*. (**f**) Oncoprint depicting alterations in genes from RTK, RAS/MAPK, and PI3K/AKT/mTOR pathways in LUAD tumors harboring *KEAP1*/*STK11* co-mutations and wild-type *KRAS* (n = 483). Alteration types include mutation, amplification, deletion, fusion, or multiple events. Over half of these tumors (n = 250) lacked any known driver alterations in these signaling pathways, suggesting alternative oncogenic mechanisms.

**Figure 4 cancers-17-01681-f004:**
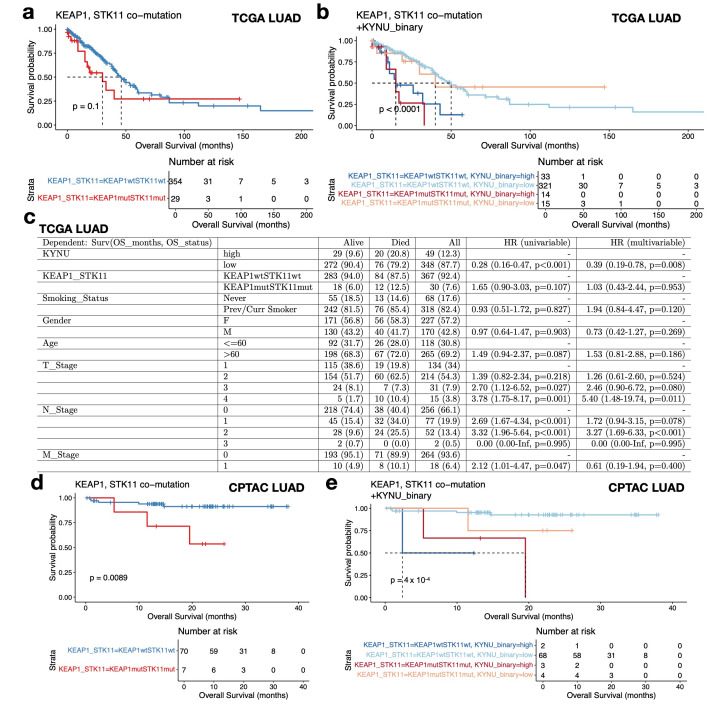
Prognostic value of *KYNU* expression independent of *KEAP1*/*STK11* co-mutations in LUAD. (**a**) Kaplan–Meier survival analysis of TCGA LUAD patients stratified by *KEAP1*/*STK11* co-mutation status. Patients with *KEAP1*/*STK11* co-mutations show shorter survival compared with wild-type patients (*p* = 0.1). (**b**) Kaplan–Meier survival analysis stratified by both *KEAP1*/*STK11* co-mutation status and *KYNU* expression status (binary high/low). High *KYNU* expression is associated with poor survival regardless of mutation status (*p* < 0.0001). Among co-mutants, *KYNU*-low patients show survival comparable to wild-type patients, while *KYNU*-high wild-type patients experience significantly worse survival. (**c**) Multivariate analysis controlling for *KEAP1*/*STK11* status, smoking status, age, gender, and TNM stage confirms *KYNU* expression as an independent predictor of poor survival in TCGA LUAD. (**d**) Kaplan–Meier survival analysis for CPTAC LUAD patients stratified by *KEAP1*/*STK11* co-mutation status. Patients with co-mutations show shorter survival compared with wild-type patients (*p* = 0.0089). (**e**) Kaplan–Meier survival analysis of CPTAC LUAD patients stratified by both *KEAP1*/*STK11* co-mutation status and *KYNU* expression (high vs. low). High *KYNU* expression predicts poor survival independent of mutation status (*p* = 0.0004), consistent with findings from TCGA.

**Figure 5 cancers-17-01681-f005:**
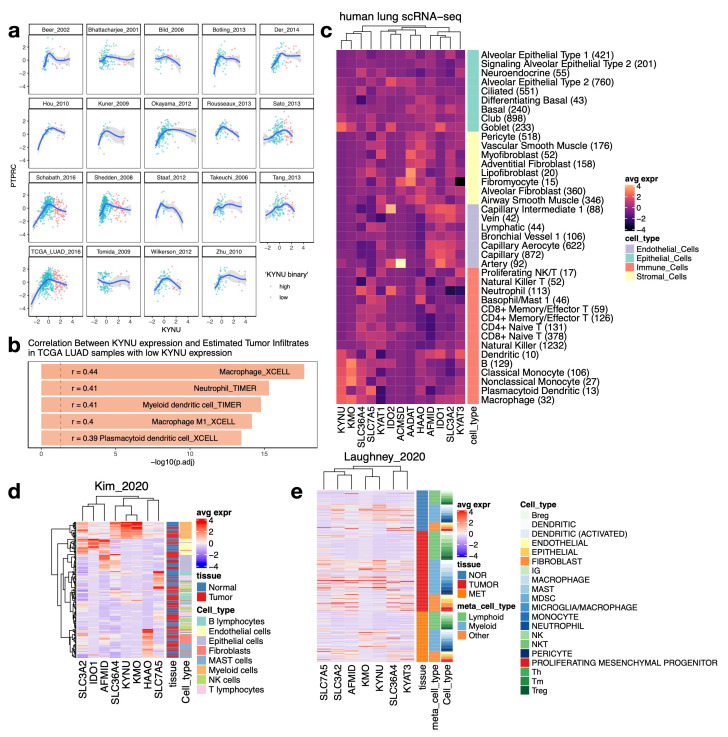
Distinct cellular sources of *KYNU* expression in LUAD tumors. (**a**) Scatterplots showing the relationship between KYNU and PTPRC (CD45) expression across 19 LUAD tumor datasets. KYNU expression was classified into high/low groups using model-based clustering. LOESS regression curves with confidence intervals highlight a biphasic relationship, where *KYNU* and *PTPRC* are positively correlated only in KYNU-low tumors. (**b**) Top five correlations between KYNU expression and immune cell infiltrates in TCGA LUAD KYNU-low tumors. Immune cell infiltration was estimated using deconvolution algorithms, showing the strongest associations with macrophages, neutrophils, and dendritic cell scores. (**c**) Single-cell RNA-seq data from the healthy human lung showing *KYNU* expression across various cell types. Myeloid lineage cells, including macrophages and plasmacytoid dendritic cells, exhibit high *KYNU* expression, while epithelial cells show minimal expression, except for goblet cells. (**d**,**e**) Single-cell RNA-seq analysis of human NSCLC tumors confirming *KYNU* expression predominantly in myeloid cells. Panels show KYNU and other tryptophan metabolism-related genes in tumor-infiltrating immune cells and cancer cells. See reference [3] (LCE) for the dataset origins in panel (**a**), [68] for the Kim_2020 dataset in panel (**d**), and [69] for the Laughney_2020 dataset in panel (**e**).

**Figure 6 cancers-17-01681-f006:**
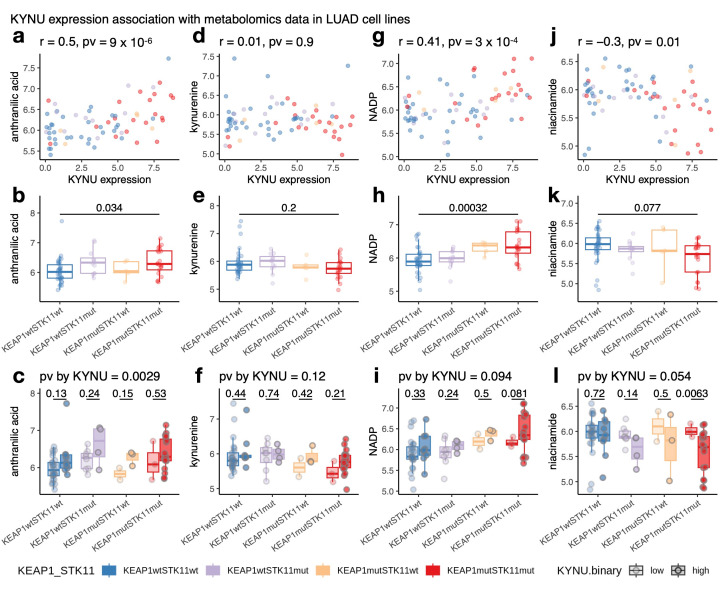
Association of *KYNU* expression with metabolomics data in LUAD cell lines. (**a**) Scatterplot showing the relationship between *KYNU* expression and anthranilic acid levels, the direct product of kynureninase activity, in LUAD cell lines. (**b**) Boxplot comparing anthranilic acid levels across *KEAP1*/*STK11* oncogenotypes, with *p*-values derived from one-way ANOVA. (**c**) Boxplots comparing metabolite levels by *KYNU* expression (high vs. low) within each *KEAP1*/*STK11* oncogenotype subgroup. T-tests were used for within-group comparisons. The *p*-values in the plot title represent metabolite differences by *KYNU* expression status from a multivariate analysis adjusting for *KEAP1*/*STK11* mutations (**d**–**l**). Similar assessments for kynurenine (**d**–**f**), NADP (**g**–**i**), and niacinamide (**j**–**l**). The color scheme used in all panels corresponds to the legend shown at the bottom of the figure.

**Figure 7 cancers-17-01681-f007:**
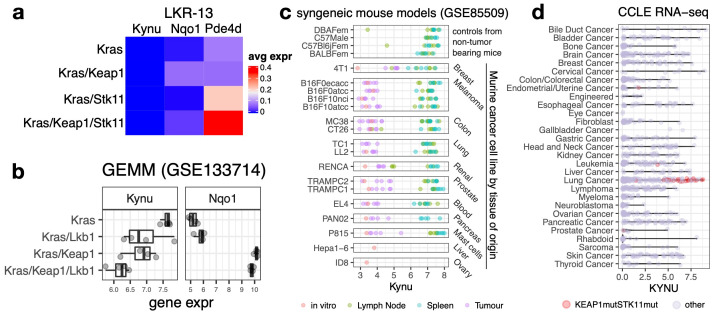
Challenges in modeling Kynu expression in preclinical LUAD systems. (**a**) Analysis of scRNA-seq data from syngeneic LKR13 mouse models with *Kras*, *Kras/Lkb1*, *Kras/Keap1*, and *Kras/Keap1/Stk11* genetic backgrounds. *Kynu* expression is absent in cancer cells across all genotypes, while *Nqo1* (Nrf2-regulated) and *Pde4d* (Lkb1-suppressed) serve as controls, showing expected expression patterns. (**b**) *Kynu* and *Nqo1* expression in autochthonous tumors from genetically engineered mouse models. *Kynu* expression is highest in Kras tumors and decreases in *Kras/Keap1* and *Kras/Stk11* mutants, with the lowest levels observed in *Kras*/*Keap1*/*Stk11* mutants—opposite to the patterns observed in human tumors. (**c**) *Kynu* expression across syngeneic mouse models of different cancer lineages. *Kynu* expression remains low in tumor tissues but is markedly higher in immune-rich tissues, such as spleen and lymph nodes. (**d**) *KYNU* expression in human cancer cell lines from the CCLE dataset. Non-LUAD cancer cell lines show robust, cancer cell-intrinsic *KYNU* expression, with upregulation independent of *KEAP1/STK11* co-mutations, suggesting alternative mechanisms of transcriptional regulation.

**Figure 8 cancers-17-01681-f008:**
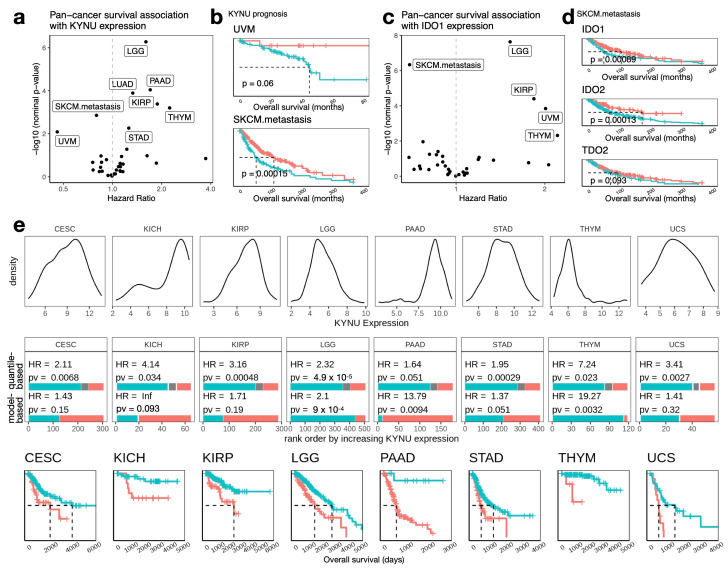
Prognostic significance of *KYNU* and kynurenine pathway genes across cancer types. (**a**) Hazard ratios (HR) and *p*-values for *KYNU* expression across multiple cancer types from univariate Cox proportional hazards regression. Cohorts with significant associations (adjusted *p* < 0.05, Benjamini–Hochberg correction) are labeled. (**b**) Kaplan–Meier survival plots for *KYNU*-high and *KYNU*-low groups in uveal melanoma (UVM) and metastatic skin cutaneous melanoma (SKCM.met). Groups were defined using model-based clustering, and *p*-values were determined by log-rank tests. (**c**) Hazard ratios and *p*-values for *IDO1* expression across cancer types were analyzed using the same approach as panel (**a**). (**d**) Kaplan–Meier plots for *IDO1*, *IDO2*, and *TDO2* expression in SKCM.met, showing their association with improved survival. (**e**) *KYNU* expression and survival associations in selected TCGA cancer cohorts. Top: *KYNU* expression distributions by cohort. Middle: Hazard ratios and *p*-values comparing *KYNU*-high and *KYNU*-low groups from Cox regression. Bottom: Kaplan–Meier survival curves for *KYNU*-high and *KYNU*-low groups defined using the dichotomization method (quantile-based or model-based clustering) that yielded the most significant *p*-value. Quantile-based dichotomization classified tumors with expression above the 80th percentile as *KYNU*-high and below the 70th percentile as *KYNU*-low. Kaplan–Meier plots show median survival (dotted lines). Blue and red lines represent low and high expression groups, respectively.

## Data Availability

All datasets analyzed in this study were obtained from publicly available repositories, including TCGA (https://portal.gdc.cancer.gov/), CCLE (https://depmap.org/portal/), CPTAC (https://proteomics.cancer.gov/programs/cptac), and Synapse (Travaglini scRNA-seq, https://www.synapse.org/#!Synapse:syn21041850). Detailed information on file versions, download dates, and preprocessing steps is provided in the Methods section. Processed data supporting the findings of this study are available from the corresponding author upon reasonable request.

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
