# Peer review of "High KYNU Expression Is Associated with Poor Prognosis, KEAP1/STK11 Mutations, and Immunosuppressive Metabolism in Patient-Derived but Not Murine Lung Adenocarcinomas"

_cancers, 2025, doi:10.3390/cancers17101681_

Round 1
Reviewer 1 Report
Comments and Suggestions for Authors
This is a substantially satisfactory submission, with little to criticise. However, I should like to see some comment on :-
- a) the identity of their target gene, given the view that there are significant differences between KYNU1 - converting kynurenine to anthranilate and KYNU2 - converting 3-HK/ to 3HAA.
How does the present work relate to either or both of these two pathways?
- b) how relevant are measurements of anthranilic acid in work such as this, in view of its synthesis and metabolism in bacteria present in the gut, tissues etc. and indeed its metabolism by those bacteria?
- c) there is much evidence that one of the most active tryptophan metabolites in the pathway is kynurenic acid, not kynurenine, and it is of course very difficult to distinguish between them since kynurenine will always be converted to kynurenic acid. Therefore the stability of kynurenine levels may not be refleccting the levels of kynurenic acid, which is also generated independently by IL4i1
Reviewer 2 Report
Comments and Suggestions for Authors
Cai et al leveraged multiple published databases to investigate KYNU expression and its correlation with outcomes and mutation status in lung adenocarcinoma. They found that high expression of KYNU leads to worse outcome independent of KEAP1/STAK11 status. And identified various sources of KYNU. They also demonstrated the prognostic value of KYNU in other cancer types. This study exemplified the strength of leveraging public data and bioinformatics tools. However, I have the following concerns about this study.
My major concern is the weak link between different sections, making it difficult to understand the story. Eg, in figure 5, they concluded “KYNU expression represents two distinct biological contexts in LUAD”. Then how did it lead to the exploration of metabolomics.
My other comments are as below
Major
- It is not clear to me how the authors integrated 23 LUAD datasets. If they were analyzed together, what methods were performed to account for batch effects? Is there were analyzed separately, how was KYNU reached? Please clarify
- I cannot find supplemental materials, which dampen my evaluation of the manuscript.
- Comparison of KYNU level with healthy donors can further underline its role as a target.
- “high KYNU expression is consistently associated with poor survival outcomes, regardless of KEAP1/STK11 405 co-mutation status (Figures 4d-e)”--- Figure 4d showing KEAP1/STK11 co-mutation is associated with worse outcome, how could this support the statement? A Cox regression with KYNU expression and mutation status should help validate.
Minor:
- Line 278-290 should be trimmed and moved tothe discussion
- Please correct all KM curves to make sure the risk table aligns with the graph.
- Line 538: The Abbreviation LUIAD has already been introduced earlier.
- Figure 8b,d, the figure legend is missing. X-axis tile should indicate whether survival is in days/weeks/months etc.
Round 2
Reviewer 2 Report
Comments and Suggestions for Authors
My comments have been addressed.